# Folding and Insertion of Transmembrane Helices at the ER

**DOI:** 10.3390/ijms222312778

**Published:** 2021-11-26

**Authors:** Paul Whitley, Brayan Grau, James C. Gumbart, Luis Martínez-Gil, Ismael Mingarro

**Affiliations:** 1Department of Biology and Biochemistry, Centre for Regenerative Medicine, University of Bath, Bath BA2 7AY, UK; bssprw@bath.ac.uk; 2Department of Biochemistry and Molecular Biology, Institute of Biotechnology and Biomedicine (BIOTECMED), Universitat de València, E-46100 Burjassot, Spain; Brayan.Grau@uv.es (B.G.); Luis.Martinez-gil@uv.es (L.M.-G.); 3School of Physics, School of Chemistry and Biochemistry, Parker H. Petit Institute for Bioengineering and Bioscience, Georgia Institute of Technology, Atlanta, GA 30332, USA; gumbart@physics.gatech.edu

**Keywords:** folding, insertion, membrane protein, translocon, ribosome, transmembrane segment

## Abstract

In eukaryotic cells, the endoplasmic reticulum (ER) is the entry point for newly synthesized proteins that are subsequently distributed to organelles of the endomembrane system. Some of these proteins are completely translocated into the lumen of the ER while others integrate stretches of amino acids into the greasy 30 Å wide interior of the ER membrane bilayer. It is generally accepted that to exist in this non-aqueous environment the majority of membrane integrated amino acids are primarily non-polar/hydrophobic and adopt an α-helical conformation. These stretches are typically around 20 amino acids long and are known as transmembrane (TM) helices. In this review, we will consider how transmembrane helices achieve membrane integration. We will address questions such as: Where do the stretches of amino acids fold into a helical conformation? What is/are the route/routes that these stretches take from synthesis at the ribosome to integration through the ER translocon? How do these stretches ‘know’ to integrate and in which orientation? How do marginally hydrophobic stretches of amino acids integrate and survive as transmembrane helices?

## 1. Introduction

The majority of, if not all, integral membrane proteins distributed throughout the endomembrane network in eukaryotic cells first assemble into the endoplasmic reticulum (ER) membrane. Following this assembly, these proteins are distributed to their intended destinations via specific trafficking pathways. The signals and machineries that direct complex endomembrane trafficking pathways are beyond the scope of this article. Thus, we will concentrate on the initial folding and assembly of proteins into the ER membrane and not their subsequent cellular distribution. 

Most integral membrane proteins require proteinaceous machinery known as an integrase for their insertion into the ER [1]. The Sec61 translocon/integrase is the primary integration machinery of the ER although there are others, notably the Get1/Get2 integrase that is responsible for tail-anchored (TA) protein integration [2,3]. A further integrase called the ER membrane protein complex (EMC) is also implicated in post-translational membrane integration of a subset of TA proteins but also seems to play additional roles in co-translational membrane protein assembly [4]. These integrases have been evolutionary conserved in eukaryotes [5] with some of them, such as Sec61 and Get1/2, having homologs in prokaryotes. In this article, we will exclusively consider membrane protein assembly mediated by the Sec61 translocon.

## 2. Structure-Function of the Translocon

The mammalian Sec61 translocon consists of a core heterotrimeric Sec61α, β, and γ complex which have 10, 1, and 1 transmembrane (TM) domains respectively [6]. In addition, numerous accessory proteins are associated with this core complex which may modulate translocon activity or provide functionality that complements its translocation and integrase function. Accessory protein complexes such as TRAP seems to be involved in membrane protein topogenesis [7,8,9] and oligosaccharyltransferase, which adds oligosaccharides to asparagine residues in the lumen of the ER (at NXS/T sequons), are commonly found associated with the translocon core and/or are present in approximately stoichiometric amounts [10,11]. Others such as signal peptidase complex, TRAM, or Sec62/63 are present in sub-stoichiometric amounts and may only associate transiently or under certain circumstances when particular substrates are present in the translocon [11]. Recently, it was discovered that the EMC is also present as a cooperative partner of the Sec61 translocon machinery during co-translational membrane-protein insertion [12]. Thus, the translocon should be considered as a dynamic rather than a well-defined complex.

The structure of the core Sec61 translocon and homologous complexes such as SecYEG of *Escherichia coli* and SecYEβ of the archaeal *Methanococcus janaschii* have been extensively studied [6]. The structures are all fundamentally similar, suggesting a common mode of action in facilitating the translocation of polypeptide chains across membranes and the integration of appropriate stretches of amino acids into the lipid bilayer. The structure/function relationship of all translocons that is most widely accepted is as follows: Looking from the top, onto the membrane the 10 TM subunit (Sec61α/SecY) is pseudosymmetrical with TM domains 1–5 and 6–10 forming the two halves in a clamshell arrangement (Figure 1A–C); A hinge region at one interface between the two halves consisting of TM5 and TM6 allows for the opening and closing (breathing) of a lateral gate between TM2b and TM7 at the opposite side of the clamshell; this breathing is proposed to allow lateral movement of polypeptides out of the translocon and into the membrane providing the integrase function. From a different view, looking into the membrane from the side (Figure 1D) there appears to be a continuous channel through the membrane that is constricted at the center of the membrane and wider at both extremities, resembling an hourglass. The constriction is lined with six hydrophobic amino acid residues [13]. Furthermore, in structures without a translocating polypeptide, there is a small ‘plug’ helix (on the lumenal/non-cytoplasmic side) blocking the continuous channel through the membrane. It is proposed that this small helix prevents the leakage of ions when the translocon is not occupied by a translocating polypeptide chain [14]. When there is a translocating or inserting polypeptide chain it is proposed that the plug helix is displaced so that it no longer blocks the continuous channel [15] (Figure 1C). Ion leakage is still prevented by the presence of the translocating polypeptide chain and the hydrophobic amino acids in the central constriction, which has been proposed to act as a greasy gasket forming a tight seal around the translocating chain [16,17]. However, the gasket role of the central constriction is seemingly dispensable as yeast expressing mutant Sec61 with the hydrophobic residues replaced with even charged residues can grow and perform translocation efficiently [18]. Regarding the β- and γ-subunits of the Sec61 translocon, the former is not essential for TM insertion, and the latter acts as a clamp that brings both halves of Sec61α together [19].

## 3. Targeting to the Translocon

In higher eukaryotes such as mammals, the majority of proteins utilizing the Sec61 translocon for entry into the ER do so in a co-translational mode [20,21,22]. It should be noted, however, that there is also a post-translational mode of targeting to the ER that is more prevalent in lower eukaryotes such as fungi that will not be considered here [22]. In co-translational mode, proteins that are to be targeted to the ER are synthesized with a hydrophobic N-terminal sequence. This sequence can either be a signal peptide (signal sequence) that is eventually cleaved from the mature protein once it reaches the ER or an uncleaved TM segment (signal anchor) which comes in two different flavors, conventional (N-terminal cytoplasmic) (Figure 2) or reverse (N-terminal ER lumen). Upon emerging from the ribosome exit tunnel, the N-terminal hydrophobic sequence on the newly synthesizing (nascent) protein is recognized by a ribonucleoprotein chaperone complex called signal recognition particle (SRP). The binding of SRP arrests translation by the ribosome and a ribosome/nascent chain/SRP complex is formed. Targeting to the ER membrane is facilitated by a specific interaction between SRP and an SRP receptor, which is integral in the ER membrane. SRP releases from the complex upon hydrolysis of GTP, reversing the arrest in translation with the ribosome/nascent chain situated on top of a translocon (Figure 2).

## 4. Getting across the Membrane

When translation resumes in the vicinity of the translocon, signal sequences/anchors have to orient themselves appropriately within the translocon. Positively charged residues positioned at one end of the hydrophobic stretch of amino acids play an important role in establishing the orientation of this initial interaction with the translocon together with other ER components, i.e., lipids [24] or potentially accessory proteins. Positive charges at the N-terminus of signal sequences and signal anchors position these types of topogenic sequences with their N-termini in the cytoplasm and direct translocation of C-terminal sequences into the lumen of the ER (N_in_-C_out_) (Figure 2). When situated at the C-terminus of a hydrophobic sequence, a C-terminus cytoplasmic orientation is favored and the N-terminus is translocated (N_out_-C_in_) [25]. It should be noted that other factors such as the length of hydrophobic stretch of amino acids or folding of previous sequence domains can affect this initial topogenic decision [26,27].

A cryo-EM structure of a ribosome/Sec61 complex in the process of translating/translocating pre-prolactin stalled at 86 residues shows the signal sequence positioned next to TM2 and TM7 in the open lateral gate of Sec61α and density in the translocon channel and ribosome tunnel are consistent with a looped configuration of the nascent chain [28]. If translation were to proceed, the loop would get longer on the lumenal side of the membrane as the polypeptide chain extends (Figure 2), at least until the signal sequence is cleaved and the translocating sequence is no longer a loop. 

It has been shown using glycosylation mapping that translocating polypeptides can span the Sec61 translocon in an extended conformation but that α-helices can also be accommodated in the translocon [29,30]. It is broadly accepted that more hydrophobic polypeptides in an α-helical conformation move from the translocon channel into the lipid bilayer more easily than less hydrophobic segments in an extended conformation that will be translocated to the ER lumen. The potential routes that topogenic signals in nascent chains take into the translocon are the topic of later sections in this article.

## 5. Integrating into the Membrane

Membrane-spanning domains are typically made up of around 20 hydrophobic amino acids in an α-helical conformation. This is sufficient to span the 30 Å hydrophobic core of a model lipid bilayer as each amino acid contributes 1.5 Å to the length of a helix. Not only does this feature of TM domains make them stable in a hydrophobic rather than aqueous environment, but it is also apparent that high hydrophobicity is a major driving force for integration [31]. There seems to be a hydrophobicity threshold above which stretches of amino acids partition into the membrane from the translocon, and below which they continue their translocation through the translocon and into the lumen of the ER. The majority of single-span membrane proteins have TM domains that have a hydrophobicity above this threshold [32] and therefore are theoretically capable of integration in an autonomous (unassisted) way. 

This all seems very straightforward. A sufficiently hydrophobic sequence will integrate while others will not. However, things are not that simple. Depending on the topology of the nascent polypeptide, inversions can occur via two sequential energetic transitions of the TM segment: first, the insertion, driven by the hydrophobic effect, and second, the inversion that has been proposed to be driven by electrostatic interactions between the nascent chain and the translocon (or associated proteins) and/or membrane lipids [33]. Furthermore, in multi-spanning proteins, at least 25% of TM helices from proteins of known structure do not reach the threshold that would allow them to integrate autonomously by a simple hydrophobic partitioning [32,34]. Low-hydrophobic segments could be helped by interacting with upstream or downstream hydrophobic segments in the nascent chain to adopt a TM disposition [35] since photocrosslinking experiments have shown that the translocon can simultaneously accommodate more than one TM helix [36,37,38] (Figure 3).

## 6. Where Does Folding of TM Helices Occur?

First, we want the reader to reflect on why TM domains of the type of integral membrane protein we are considering in this review are exclusively α-helical. The answer is that a regular pattern of hydrogen bonding between carbonyl oxygens and amide hydrogens in peptide bonds in an α-helix secondary structure ensures maximal hydrophobicity by effectively neutralizing polar groups in the peptide bonds. It has been calculated that a stretch of hydrophobic amino acids in an extended conformation could not reach the hydrophobic threshold to partition through the lateral gate into the lipid environment [39,40,41,42]. Thus, we assume that hydrophobic stretches of amino acids must be in an α-helical conformation to partition into the lipid environment in a thermodynamically favorable way. Hydrophobic stretches of amino acids may fold into α-helices in the translocon tunnel as they enter. However, it is also feasible that the secondary structure is formed before entering the translocon (Figure 2).

During co-translational membrane insertion secondary structures can be acquired in the ribosome specifically in the ribosome exit tunnel. This tunnel accommodates nascent peptides from the ribosomal peptidyl transferase site (P-site) to the ribosomal exit site, providing a protective shield. The first evidence for folding in the ribosome before insertion came more than two decades ago, indicating that the constrained environment of the ribosome-translocon complex has an active role in the propensity of some nascent chains to acquire an extended or a more compact conformation [29]. Furthermore, the authors determined that this compaction was acquired co-translationally in an amino acid-dependent manner. A few years later, it was discovered that the ribosome exit tunnel itself provides enough space to allow the folding of secondary structures as α-helices as Johnson and colleagues demonstrated attaching fluorescent FRET partners at different positions in nascent peptides. The results indicated that folding of TM sequences in an α-helix-like structure was induced and stabilized far inside the ribosome exit tunnel and close to the ribosomal P-site, whereas a nascent secretory protein remained in an extended conformation within the ribosome exit tunnel [43]. These results were further supported by biochemical assays based on pegylation of cysteine reporters [44]. These studies also established that there are seemingly different folding zones within the ribosome exit tunnel where the secondary structure formation can occur, as alanine-replaced peptide segments used for the pegylation experiments folded more compactly when located near the P-site than when located more distally in the tunnel [45]. 

Following these biochemical studies where compaction could be inferred, atomic structures of α-helical nascent polypeptide chains visualized within the ribosome exit tunnel were quick to follow. Utilizing single-particle cryo-electron microscopy reconstitutions of eukaryotic 80S ribosomes containing nascent chains with an α-helical propensity, Beckmann and collaborators visualized helix density inside the tunnel as well as interaction sites with the tunnel wall components [46]. 

It is thought that α-helix formation occurs first wherever it is possible and that the ribosome may be selective in the types of sequences that it allows to form secondary structures. Using glycosylation mapping and molecular dynamics (MD) simulations [47] it was shown that in combination, hydrophobicity, helical propensity, and length of hydrophobic stretches of amino acids are major determinants for α-helix adoption within the ribosome exit tunnel (Figure 2). In these experiments, nascent chains harboring α-helix sequences from TM segments were able to fold inside the ribosome exit tunnel whereas those with α-helical sequences of soluble proteins were not. 

It is, however, not only secondary structures that can be adopted inside the ribosome tunnel. Small domains with a tertiary structure such as zinc-finger domains have been observed to fold close to the ribosome exit site, also called the vestibule, where the ribosome exit tunnel widens substantially [48], with this folding accelerated and stabilized by the tunnel [49]. Furthermore, helical TM hairpin formation is also possible in the vestibule of the ribosome exit tunnel [50]. Currently, we think of the ribosome/translocon complex as specialized chaperone-like machinery implicated in the formation of secondary or tertiary structure as a platform to overcome the huge energy barrier required to insert unfolded hydrophobic sequences and to prevent their exposure on the cytosolic side of the ER membrane [36,47].

The implication of the ribosome in promoting the secondary structure of TM domains raises a question, ‘could this be an evolutionarily conserved function of ribosomes? Presumably, the very first TM proteins had to insert into membranes without the assistance of translocon machinery. There are numerous examples of α-helical membrane proteins that can insert into simplified biomimetic systems without translocon assistance [51,52,53]. Recently proteorhodopsin has been shown to spontaneously integrate into a simple lipid membrane in the absence of chaperones or a translocon [54]. Other evidence of the pre-translocon era is that yeast mitochondrial ribosomes permanently attach to the mitochondrial inner membrane which lacks the translocon [55,56,57]. Nevertheless, other components tethering translating ribosomes to the mitochondrial inner membrane have been identified [55]. The permanent docking of these ribosomes facilitates the insertion of membrane proteins encoded by mitochondrial DNA efficiently. Therefore, not only the ribosome but also the recruitment of translating ribosomes to membranes seem to be crucial to a successful insertion and folding process. All these examples highlight the importance of the ribosome in the process of membrane insertion, however, we should be mindful that it is not only TM domains and ribosomes that have co-evolved. There is evidence that extant biological membranes have arisen as a consequence of a lipid bilayer and membrane protein co-evolution process [58]. Altogether, it is highly probable that the ribosome’s ability to assist in α-helix folding is not a coincidence exploited by membrane proteins for rapid structure acquisition before membrane insertion; rather it seems likely it is the result of a fine-tuned co-evolution process that continues to this day.

## 7. Route Into, Through, and Out of the Translocon

In addition to its insertase activity, the translocon is responsible for the secretion of proteins to the ER lumen. The insertion or secretion of proteins can occur both co-translationally or post-translationally depending on the presence of an N-terminal signal sequence (SS). The SS is characterized by a hydrophobic core, often the first TM segment for membrane proteins, flanked by positively charged amino acid residues and polar but uncharged residues at the N- and C-termini, respectively [59]. If the SS is present, once the ribosome is brought to the membrane, it engages with the Sec61 translocon while the nascent chain is still being translated (Figure 2). As mentioned above, the presence of positively charged residues at the N-terminus of the SS causes inversion of its topology, aided through its interactions with lipids and possibly nearby ribosomal RNA [23,33,60]. 

Within the ribosome/translocon complex the cytoplasmic entrance of the Sec61α channel has a diameter of 20–25 Å [13] (Figure 1D), which nicely matches with the diameter of the ribosome exit tunnel vestibule [61], allowing ribosome-acquired helices to enter into the aqueous translocon channel [62]. When the nascent chain arrives at the laterally closed translocon, it has been demonstrated that the presence of the SS triggers channel opening of the translocon at the lateral gate, which is also linked to the partitioning of TM α-helical structures of membrane proteins into the lipid phase [13,63]. Subsequent hydrophobic regions of the nascent chains will be exposed to the lipid phase and can partition into it as experimental data, structures, and simulations have revealed [64,65,66,67,68] (Figure 3). Recent single-molecule FRET experiments indicate that the lateral gate is highly dynamic even in the absence of a membrane-inserting SS or TM segment, although binding of the ribosome and insertion of TM segments increases the probability of the open state [69]. Indeed, earlier cryo-tomography and fluorescence studies also revealed that binding of the ribosome induces channel opening [70,71].

As noted earlier, for multi-pass membrane proteins, the topology is typically established by the first TM, which adopts either an N_in_-C_out_ (N-terminus in the cytoplasm and C-terminus in the ER lumen) or an N_out_-C_in_ topology in the translocon. Each subsequent TM segment adopts the opposite topology of the previous one as it enters the membrane. Recent evidence has implicated the EMC in helping to establish the topology of the first TM of many GPCRs and likely other proteins [12,72]. The translocon’s lateral gate allows direct sampling of the membrane environment by the TM [73], possibly from the moment of its encounter with the translocon [39,74]. The code for insertion, i.e., what sequences partition to the membrane vs. remain in the channel, has been determined in exquisite detail with extensive, elegant experiments from von Heijne, White, and colleagues in the mid-2000s [31,32]. Nonetheless, in addition to thermodynamics, a role for kinetics has also been found from both simulations [65,75,76] and experiments [18,77,78]. In particular, slowing down the rate of protein synthesis can increase the probability of membrane insertion.

While most membrane proteins have a fixed topology that alternates from one TM to the next, some defy this straightforward expectation and are known as “dual-topology proteins”. These proteins can be found in roughly equal numbers in an N_in_-C_out_ or an N_out_-C_in_ topology [79]. In particular, the topology of EmrE, a canonical example of a dual-topology protein, has been found to be very sensitive to the presence of charges, not just at its N-terminus but in any of its loops and even at its C-terminus [80]. This surprisingly suggests that even after synthesis is completed, dual-topology proteins can exist in a dynamic equilibrium where TM helices flip in and out of the membrane, probably in the vicinity of the translocon [81]. Coarse-grained simulations indicated this unusual ability arises from a lack of full integration of some TMs until well after the completion of synthesis [82].

## 8. Exploring the Limits of TM Domain Insertion

We have previously described how TM helices are proposed to partition from the translocon into the non-polar core of the lipid bilayer driven by hydrophobicity with, the limits for the insertion of TM segments being explored using computationally designed segments with naturally occurring amino acid distributions [83]. However, features found in naturally occurring TM domains such as limited length, charged residues, or a low hydrophobicity profile challenge this rather simplistic view of what a TM segment is.

Given that exposure of hydrophobic groups in proteins and lipids to water is highly unfavorable, membrane proteins tend to minimize their free energy by maximizing the match between the hydrophobic width of the bilayer and the length of a TM segment, a phenomenon called hydrophobic matching [84,85]. Indeed, the average length of a TM segment is 24 ± 5.6 residues long (36 Å in a 3,6 α-helix) [86], while for instance, the width of the hydrophobic core of an ER hepatocyte membrane is close to 38 Å [87,88,89]. A mismatch in length between the hydrophobic section of a membrane-spanning protein and the bilayer in which it is located results in lipid and peptide rearrangements to compensate [90]. Ultimately a major hydrophobic mismatch might prevent insertion into the membrane, but how much is too much?

The minimum hydrophobic length necessary to form a TM segment in lipid bilayers has been investigated using short hydrophobic peptides in dioleoylphosphatidylcholine (DOPC) and 1,2-dierucoyl-*sn*-glycero-3-phosphocholine (DeuPC, a shorter lipid) vesicles [91]. Peptides composed of Leu residues were compared to sequences of the same length containing alternating Leu and Ala residues (which have a hydrophobicity typical of natural TM helices) [91]. The authors observed that peptides composed exclusively of Leu residues were able to adopt a TM disposition with just 11–12 residues. In this case, the bilayer width exceeded the hydrophobic length of the peptide by ~11–12 Å. For the alternating Leu/Ala sequence 13 residues representing a negative mismatch of ~9 Å were required. The minor differences indicate that the minimum length necessary to form a TM segment is only modestly hydrophobicity-dependent, at least for the sequences tested in this study.

In vitro expression and MD simulations have also been used to systematically examine the insertion efficiency of TM segments consisting primarily of leucine residues [92,93]. Depending on the flanking residues the minimum length to achieve a ~100% insertion efficiency varied from 12, with Gly, Asn, or Asp rich flanking sequences, to 10 residues when Lys was used as a flanking residue. The MD simulations suggest that the insertion efficiency of these sequences is determined primarily by the energetic cost of distorting the bilayer in the vicinity of a short TM segment. The presence of Lys residues flanking the hydrophobic core can reduce the energetic cost by extensive hydrogen bonding with water and lipid phosphate groups (snorkeling) and by partial backbone unfolding. The unfolding is stabilized in the simulation by water molecules entering the bilayer along the peptide backbone.

The studies cited above utilized model hydrophobic sequences composed of only a few different kinds of amino acids. However, TM segments in native membrane proteins vary significantly not only in length but also in composition, revealing more complex scenarios [86]. A computational analysis of the composition and location of amino acids in TM helices found, in membrane proteins of known structure, a strong reverse correlation between the composition/overall hydrophobicity and the required length for their insertion in the lipid bilayer [83]. These results were in accordance with in vitro studies, in which the length dependence of a TM segment (varying from 10 to 25) strongly depends on its amino acid composition [32]. Furthermore, the analysis of naturally occurring residues in TM segments put the focus on the importance of residue positioning, particularly Pro and charged residues. 

At the other extreme, the longest TM segment found in naturally occurring membrane proteins is close to 40 residues long [86]. Long TM segments usually adapt their hydrophobic length to the lipid membrane by tilting [86,94]. Accordingly, tilting of long hydrophobic sequences should occur before their insertion in the lipid bilayer, that is, before or during its partition from the translocon into the membrane. An extensively long TM segment might not be easily accommodated within the translocon in its tilted disposition which could prevent its insertion as a single TM domain. Interestingly, a 40-residue long hydrophobic sequence could potentially span a membrane twice in a helical hairpin conformation. The transformation from a long TM domain into a helical hairpin depends primarily on the presence of turn propensity residue(s) in the middle of the sequence both in natural and polyLeu sequences [95,96]. The probability of possessing one of these residues increases tremendously as sequence length extends beyond the 40-residue mark. Increasing the number of residues beyond 40 will facilitate the introduction of a turn propensity residue while maintaining the minimum distance to cross the membrane twice. 

Another challenge to the idealized view of a TM segment is the presence of charged residues. The prevalence of these residues within the buried sections of membrane proteins is very low, as expected based on their polarity [86]. However, sequence analysis of membrane protein databases shows that ionizable amino acid residues are indeed present in TM domains, often with functional and/or structural roles [86,97]. How then, are TM domains containing charged residues inserted into the hydrophobic core of the lipid bilayer? The strong hydrophobic contribution of the abundant non-polar residues in a TM segment might be greater than the energy penalty of introducing a charge into the membrane. This negative ΔΔG should be enough to promote the insertion. Interestingly, computer simulation studies suggest that the transfer of four leucines from water to the bilayer interior could be sufficient to compensate for the transfer of a cationic residue [98]. Experiments have also corroborated these predictions [99]. The position within the membrane and the polarity of the residue is also a factor that needs to be taken into consideration when determining the energy necessary for their insertion. Residues close to the water lipid interface can be hydrated due to the presence of “water defects” that reduce the insertion penalty. The closer the polar residue is to the center of the bilayer the larger (and increasingly unfavorable) the water defects become. 

The presence of water molecules within the lipid bilayer negatively affects membrane integrity. Interestingly, destabilization by the presence of water molecules has been exploited by some proteins to permeabilize the membrane. Water defect-inducing residues, that is, charged and polar residues surrounded by hydrophobic amino acids, are indeed found in pore-forming proteins [100,101,102]. Importantly, once in the membrane, polar residues strongly influence the folding or association of integral membrane proteins [97,103,104,105] and their activity [106,107].

The presence of polar residues within TM segments cannot always be explained by the large hydrophobicity of the surrounding amino acids compensating for the energy penalty of their membrane integration. As mentioned previously, around 25% of TM segments do not reach the hydrophobicity threshold required for autonomous partitioning into the membrane. In these cases, other forces should facilitate the localization of polar amino acids in the hydrophobic environment of the lipid bilayer. Interaction between TM segments within the translocon (Figure 3) provides an opportunity for polarity masking (see Section 5). Although rare, it has been demonstrated that polar interactions between neighboring helices can facilitate insertion [95,108]. Since these interactions are necessary for the partitioning into the lipid bilayer, a cooperative insertion of the TM segments is predicted which requires the presence at the same time of, at least, two helices in the translocon [109] (Figure 3). Similarly, it has been suggested that salt-bridge formation between residues located on the same face of a single TM domain may reduce the free energy of membrane partitioning.

Of note, in a low dielectric constant environment such as the membrane core, the force of an electrostatic interaction increases tremendously, thus creating very stable associations [103,104]. It has been shown that hydrogen bonding between TM segments gives stronger associations than the packing of surfaces in glycophorin A helices driven by the GxxxG interaction domain [104,110].

In channel-forming proteins, such as aquaporins (Figure 4), the presence of polar/charged residues close to the membrane hydrophobic core is not explained either by the strong hydrophobicity of the accompanying lipid-facing residues or by electrostatic interactions with other residues. Membrane channels create an amphipathic environment. Some of the residues in them (the apolar ones), regardless of the depth they are found within the membrane, will be facing the hydrophobic core of the membrane while others (the polar/charged residues) will be exposed to the water-filled tunnel of the channel (Figure 4). The nature of the residues in those two different situations varies accordingly, so the overall amino acid composition of the helices that constitute these channels resemble that of interfacial (amphipathic) helices where one side of the helix is polar while the other is filled with hydrophobic residues. In this case, there is no energy penalty for the inclusion of a charged residue in a TM domain as long as it is lining the aqueous pore of the channel. However, since the amphipathic environment is created by the channel itself, the assembly (or partial formation) of its tertiary structure must presumably involve some co-operativity between TM domains or require chaperones to avoid exposure of the hydrophilic residues to the lipids’ hydrocarbon chains. Insertion of multiple helices at once represents a challenge for integrases such as the translocon and associated components that will probably need to expand capacity to accommodate helical bundles. In the case of aquaporin 1 (AQP1) the second TM domain of six is fully translocated into the ER lumen and only adopts a transmembrane orientation after TM4 has been synthesized [111]. Membrane insertion of TM2 also requires a 180 degree flip of TM3 in the membrane presumably, but not necessarily, facilitated by the Sec61 translocon [112]. Such TM domain gymnastics during membrane protein assembly further highlight the potential flexibility of the Sec61 translocon in facilitating different insertion modalities.

## 9. Concluding Remarks

Tremendous progress has been made over the last two decades in the field of membrane protein insertion and folding, ranging from biochemical and structural data on the acquisition of secondary structure in nascent chains to the quantitative understanding of the energetic forces for sequence-dependent membrane insertion through the ER translocon and the structures of translocons engaged in nascent chain integration. This does not mean that we have a complete understanding of all aspects of membrane protein assembly and there are still challenges for the field. For example, it will be challenging to explore the dynamics and mechanisms of recruitment of translocon-associated proteins. Thus, it is difficult to envisage how proteins such as TRAM or other accessory components in the ER, which are present in sub-stoichiometric amounts can be present only at those translocons where its suggested chaperone function is needed [113]. Deducing detailed pathways of insertion of marginally hydrophobic TM domains such as TM2 of AQP1 remains another challenge. We hope that the development of new and existing technologies that can answer these remaining questions regarding membrane protein insertion and assembly is not too distant on the horizon. In recent years, new developments in cryo-EM sample preparation and data acquisition have been very fruitful in determining structures of integrase protein complexes [114] and, hopefully, they will be able to shed light on complete dynamics of the insertion process by determining intermediate states of membrane protein assembly through the aforementioned complexes. The increase in experimental data and computational power, in addition to the design of user-friendly interfaces, make MD simulations well positioned to play a very important role in the coming years in the understanding of membrane protein insertion. Simulations of ribosome–translocon complexes on the ns-μs time scale have been possible for a decade [65,115], yet full insertion and maturation take place over seconds. Coarse-grained simulations make physiologically realistic time scales achievable, albeit at the cost of detail [67]. Balancing the need for atomic resolution with that for long time scales will be an ongoing effort, made more challenging by the increasing number of players characterized structurally in integrase complexes.

## Figures and Tables

**Figure 1 ijms-22-12778-f001:**
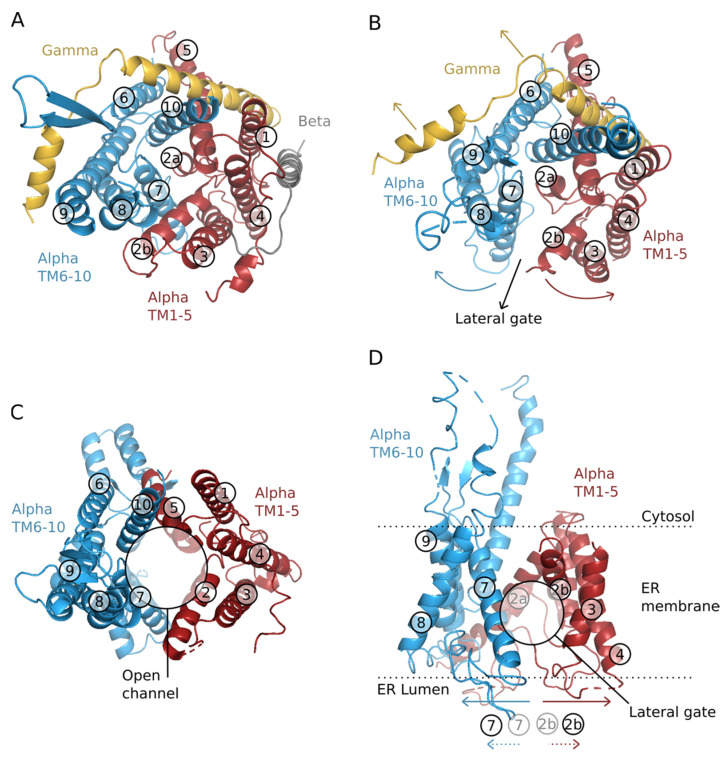
Structure of the translocon. Structure-based cartoon representations of the translocon. All TM segments of Sec61α are colored (red and blue for each half) except for the β-subunits and γ-subunits, which are shown in gray and yellow respectively. All TM segments are numbered for easy comparison between the open and closed structures. (**A**) Top view of the traslocon at a resting/close position from *M. jannaschii* (PDB: 1RHZ). (**B**) Top view of the partially open structure of the translocon from *P. furiosus* (PDB: 3MP7). The colored arrows in red and blue indicate the helix displacements required for the widening of the channel and opening of the lateral gate. The black arrow shows the lateral gate exit pathway of a TM segment from the interior of the channel into the membrane. (**C**) Top view of the translocon with the plug (TM 2a) open from *Saccharomyces cerevisiae* (PDB: 7KAL). The open channel facilitates translocation into the ER lumen of secreted and TM proteins segments. (**D**) Lateral view of the partially open structure of the translocon from *P. furiosus* (PDB: 3MP7). The colored arrows (red and blue) indicate the helix displacements required for the widening of the channel and opening of the lateral gate. The dotted colored arrows at the base of the panel indicate the movement of helices 2b and 7 (compared to a close state, panel A). The limits of the membrane are shown with a dotted line, data from the OPM database (https://opm.phar.umich.edu/ (accessed on 18 November 2021)).

**Figure 2 ijms-22-12778-f002:**
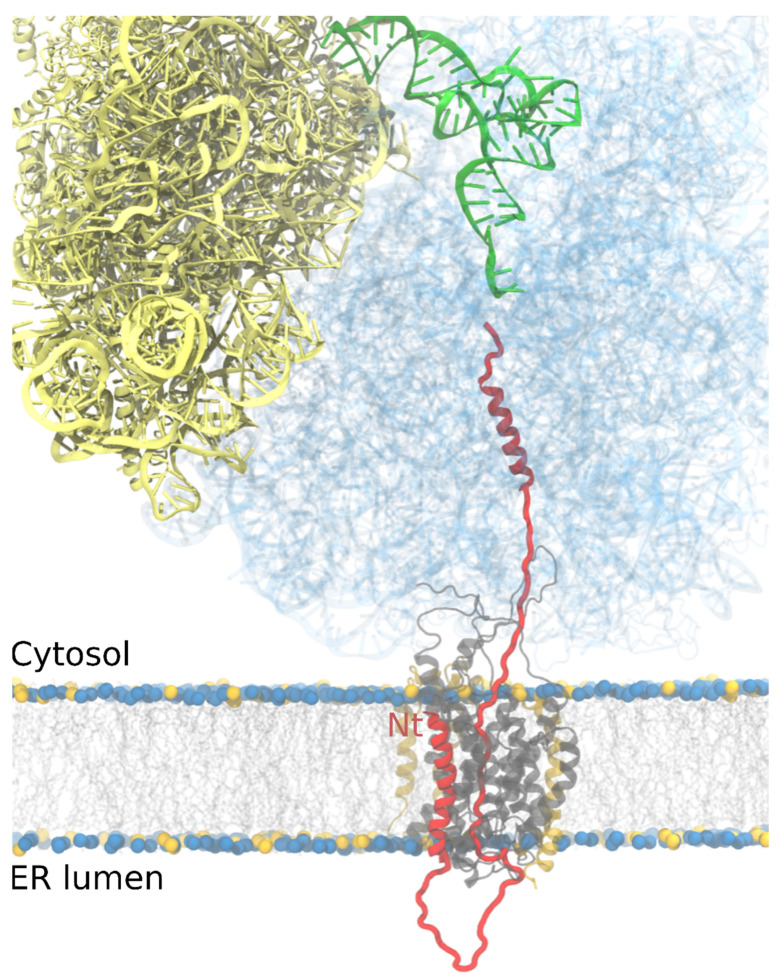
Pathway of a nascent protein. The ribosome is shown in yellow (small subunit) and light blue (large subunit, transparent). A P-site tRNA is shown in green. The nascent chain (red) traverses the ribosome’s exit tunnel before encountering the membrane-bound translocon complex (transparent orange and grey). The nascent chain has been shown to form a secondary structure in the translocon as well as at the ribosome’s exit tunnel. Most of the structure shown is taken from PDB 4V6M [23], although the nascent-chain helix inside the exit tunnel is modeled.

**Figure 3 ijms-22-12778-f003:**
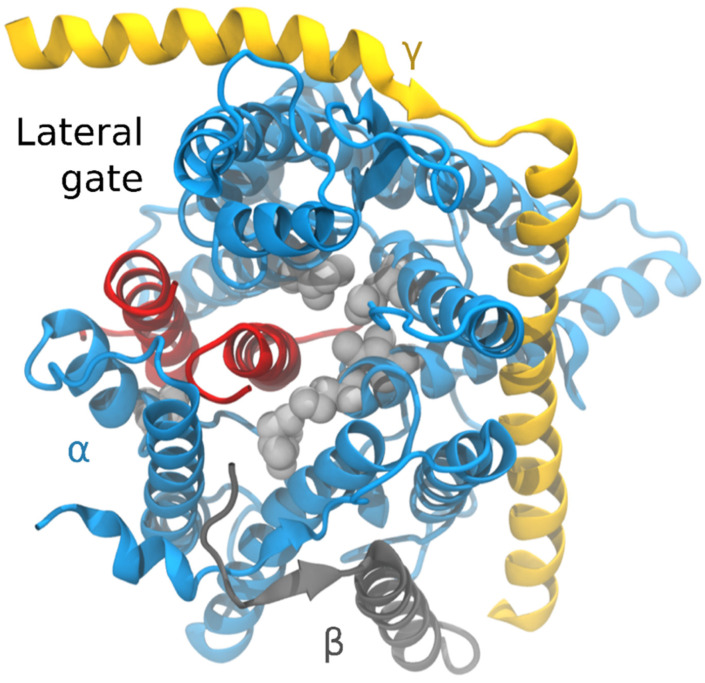
The translocon is shown from the cytoplasmic side in light blue (Sec61α), orange (Sec61γ), and grey (Sec61β). Two helices of a nascent chain are shown in red, one residing in the translocon channel and one at the open lateral gate. Pore ring residues are shown in a white space-filling representation. The structure was modeled based on MD simulations of PDB 1RHZ [13].

**Figure 4 ijms-22-12778-f004:**
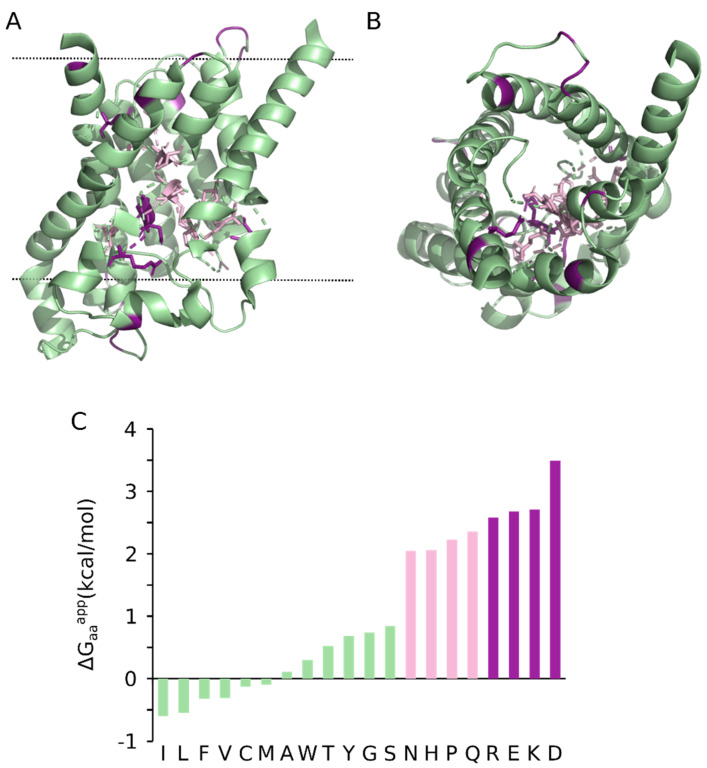
Crystal structure of human aquaporin 10. Cartoon representation of the human aquaporin 10 (PDB:6F7H), (**A**) lateral view, (**B**) top view. Residues are colored based on their membrane insertion propensity: Residues with a low insertion propensity are shown in deep purple and pink. Residues with a mild and high insertion propensity are depicted in green. The side chain of polar and charged residues within the protein’s pore are shown in a stick representation. In panel A the limits of the membrane are shown with a dotted line, data from the OPM database. (**C**) Apparent free energy for the insertion in biological membranes of the 20 natural amino acids. Data from [31].

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
