# Peer review of "Folding and Insertion of Transmembrane Helices at the ER"

_ijms, 2021, doi:10.3390/ijms222312778_

Round 1
Reviewer 1 Report
In this article, authors present a review of literature on understanding folding and insertion of primarily helical proteins where large biomolecular machinery including translocon and ribosome are involved. The review is divided into subsections on various stages of the overall process and what is currently known. Some questions are posed intermittently and speculations are provided more so than a concrete conclusion or understanding. Some exceptions to known rules of hydrophobic mismatch or partitioning of residues into a hydrophobic environment are also provided. The reading is assisted by a few minimal figures. Overall, it is a publishable short review, but authors should consider some suggestions below for revising the manuscript:
- Section 2 on page 2 discusses key structural details of the translocon but no figures is provided to assist in understanding descriptions on helices, lumen, etc. it may be helpful to have a good structural details figure here.
- While details known from existing studies are outlined, an attempt is not made to synthesize higher level understanding or future directions for experimental or computational studies. Something along these lines should be included.
- I did not find any outline of currently unknown major questions. Having a specific section toward the end of the manuscript would be helpful to draw attention for future work that this review may inspire.
- Key terminologies could be assembled in a tabulated form or abbreviations list etc.
- Although review is not specifically focused on biomolecular simulations, clearly these simulations have had an impact on our mechanistic understanding of these systems. Are there lessons in existing simulation studies or challenges for future simulation studies? For example, how could one treat ribosome, translocon, lipid bilayer to create an all-atom model and still realistically simulate to probe mechanistic questions—sounds a bit wishful thinking if anything could be learned. Alternate could be to coarsen the system or leave out most complex components and focus on simpler (still challenging) yet tractable. Whatever the lines of reasoning, some authors appear to have experience in the simulation area and could add further discussions.
Author Response
We thank the reviewer suggestions that have undoubtably improved our review as detailed in each point.
- Section 2 on page 2 discusses key structural details of the translocon but no figures is provided to assist in understanding descriptions on helices, lumen, etc. it may be helpful to have a good structural details figure here.
We have included a new Figure 1 (page 3) to describe in detail the structure of the translocon.
2. While details known from existing studies are outlined, an attempt is not made to synthesize higher level understanding or future directions for experimental or computational studies. Something along these lines should be included.
3. I did not find any outline of currently unknown major questions. Having a specific section toward the end of the manuscript would be helpful to draw attention for future work that this review may inspire.
These points 2 and 3 have been tackled at Section 9 (pages 13 and 14).
4. Key terminologies could be assembled in a tabulated form or abbreviations list etc.
Abbreviation list has been added (page 1).
5. Although review is not specifically focused on biomolecular simulations, clearly these simulations have had an impact on our mechanistic understanding of these systems. Are there lessons in existing simulation studies or challenges for future simulation studies? For example, how could one treat ribosome, translocon, lipid bilayer to create an all-atom model and still realistically simulate to probe mechanistic questions—sounds a bit wishful thinking if anything could be learned. Alternate could be to coarsen the system or leave out most complex components and focus on simpler (still challenging) yet tractable. Whatever the lines of reasoning, some authors appear to have experience in the simulation area and could add further discussions.
We have included a coarse-grain/atomistic balance view for future MD simulations at the concluding remarks on page 14.
Reviewer 2 Report
In this review, Whitley et al. summarize 3 decades of progress on how hydrophobic transmembrane sequences get inserted into the ER membrane. The MS is comprehensive and written clearly, and it nicely puts the recent progress in the context of what we know about TM insertion. The field would certainly benefit from this current summary. I have only minor comments:
- The EMC has recently been discovered as quite a major and versatile machinery that inserts TM helices into the ER membrane, with quite rich recent literature. Even though this MS mainly concentrates on the Sec translocon, , it would be appropriate to mention the EMC in a few more places. For example, in the paragraphs that mention Tail-Anchored proteins, and accessory subunits of the Sec translocon.
- The paragraph describing the structure of the translocon deserves its own figure where the different structural elements of the translocon can be highlighted.
- The paragraph starting in line 85 could benefit from some references.
- The figures of structures of the translocon are presumably taken from published structures. These should clearly be indicated with a reference and a PDB number.
- In the discussion of channel-forming proteins and aquaporins (lines 405-422), the authors present a view where the individual TMs cannot insert and the protein may need to acquire its tertiary fold before insertion. This view is inconsistent with studies from the Skach lab on aquaporin topology maturation, which show that some of the TMs in the protein insert on their own, followed by posttranslational topology reorientation and insertion of hydrophilic TMs.
Author Response
We thank the reviewer for her/his encouraging comments and included its suggestions on our revised manuscript.
- The EMC has recently been discovered as quite a major and versatile machinery that inserts TM helices into the ER membrane, with quite rich recent literature. Even though this MS mainly concentrates on the Sec translocon, , it would be appropriate to mention the EMC in a few more places. For example, in the paragraphs that mention Tail-Anchored proteins, and accessory subunits of the Sec translocon.
We have added new sentences emphasizing the relevance of the EMC complex both for TA-protein insertion and as translocon accessory component, with citing the recent work (page 2, lines 41-44 and 59-61).
2. The paragraph describing the structure of the translocon deserves its own figure where the different structural elements of the translocon can be highlighted.
New Figure 1 has been added (page 3).
3. The paragraph starting in line 85 could benefit from some references.
We have added the appropriate references (page 4, line 115).
4. The figures of structures of the translocon are presumably taken from published structures. These should clearly be indicated with a reference and a PDB number.
PDB codes have been added in all the figures.
5. In the discussion of channel-forming proteins and aquaporins (lines 405-422), the authors present a view where the individual TMs cannot insert and the protein may need to acquire its tertiary fold before insertion. This view is inconsistent with studies from the Skach lab on aquaporin topology maturation, which show that some of the TMs in the protein insert on their own, followed by posttranslational topology reorientation and insertion of hydrophilic TMs.
We apologize for not being clear enough about the ‘gymnastics’ followed by aquaporins (and other membrane proteins) while integrating into the membrane. We now have emphasized this issue using as examples the more relevant papers from Skach lab.